# Minimal Criteria for Saying a Neural Network Has Learned a World Model

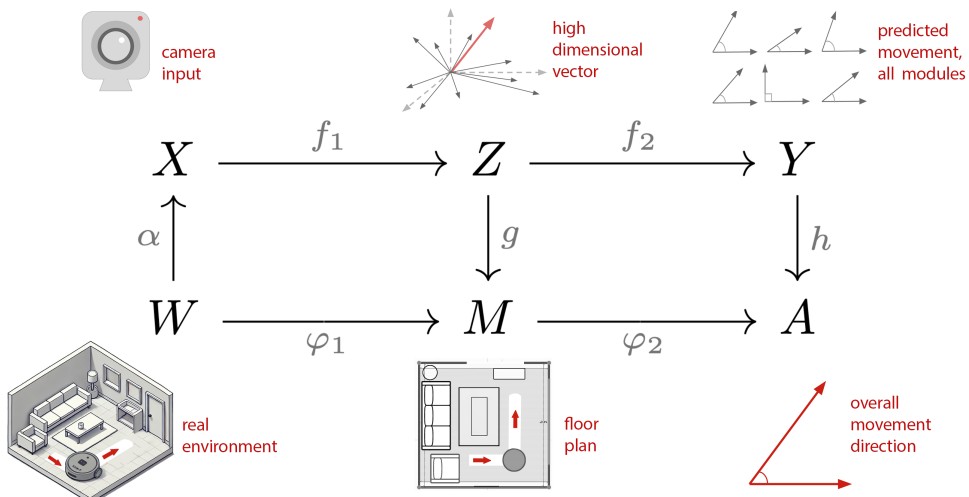

Figure 1: An example of a world model in a hypothetical neural network that controls a robot vacuum cleaner. The world $W$ is the state of a real living room, and the "world model" $M$ is the simplified state of the robot in a floorplan of the room—the result of applying a modeling function $\varphi$ to the world. The state of the world model is sufficient to determine the robot's motion in the output space $A$. Meanwhile, the world is observed by a camera (functon $\alpha$) which produces input in the space $X$ to the control network. The network computes a function $f$ to a space $Y$ of outputs to the robot's motors. We say the network uses a the "world model" $M$ if we can factor $f$ through an intermediate space $Z$, and find functions $g$ and $h$ that make the diagram commute.

## Abstract

We propose a set of precise criteria for saying a neural net learns and uses a "world model." The goal is to give an operational meaning to terms that are often used informally, in order to provide a common language for experimental investigation. We focus specifically on the idea of representing a latent "state space" of the world, leaving modeling the effect of actions to future work. Our definition is based on ideas from the linear probing literature, and formalizes the notion of a computation that factors through a representation of the data generation process. An essential addition to the definition is a set of conditions to check that such a "world model" is not a trivial consequence of the neural net's data or task.

## 1 Introduction

When a neural network is trained to make a prediction, what is it really learning? One plausible view is that the network is superficially mimicking its training data (Bender et al., 2021; LeCun, 2022). Others have suggested powerful neural networks may do something more than fragile memorization, instead building "world models" of the processes that generate their input data (see the review in Mitchell (2023)). Aside from its intrinsic interest, the question of whether neural networks model the world in any significant way connects to a wide range of issues: interface design (Viégas & Wattenberg,

2023), safety and reliability, and even basic questions about understanding and meaning (Bender & Koller, 2020).

A complication in the debate is that key terms are used informally and with different meanings. The authors have seen heated conversations where it's not clear what people are actually arguing about. Given the subtlety of the issues, it would be helpful to agree on definitions. As a baseline, we seek to move beyond vague notions such as "common sense" or "internal models of how the world works" (LeCun, 2022). Beyond precision, it would be ideal to have a set of practical criteria that can be tested experimentally.

The goal of this note is to develop criteria for a "world model" that are broadly applicable, scientific, and nontrivial. That is, a good definition should be generally consistent with the spirit of existing informal usage, lend itself to empirical experimentation, all while being non-vacuous. One might think this last criterion should go without saying, but it turns out to be a slippery point. Creating a nontrivial definition of world model turns out to be surprisingly difficult; making these difficulties explicit is an essential part of this work. Indeed, specifying what we mean by "nontrivial" leads naturally to defining supporting concepts such as what it means for a world model to be "learned," "causal," and "emergent."

The mathematical idea behind our definition is to think of a world model as a kind of homomorphic image of whatever system or process generates input data. In particular, we will say that an internal state of a network "contains" a world model if there exists some intrinsically simple (e.g., linear) function mapping the network state to this homomorphic image. A major inspiration for our definition is the large literature on linear probing. In fact, much of this paper may be seen as a reframing of ideas in Belinkov (2022). In the terms of Andreas (2024): our definition may be seen as operationalizing the notion of a "map-type" model.

In keeping with the goal of a minimal set of criteria, we don't require that a world model involve any kind of ability to simulate the results of actions; it's enough to represent the current state of the world in some nontrivial way. In conventional reinforcement-learning terms, we are focusing on what it means to represent a state space, rather than the results of actions on states. While it would be extremely useful to have concrete, testable criteria for whether a neural net does some sort of simulation under the hood, as for example described in Andreas (2024), that is beyond the scope of this work. We believe that the present set of criteria can already be useful, and in addition hope they can serve as a foundation for future investigations into action representations.

An important feature of this definition is that it focuses attention on the internals of a network, rather than behavior. In particular, we do not assume that a system with perfect accuracy on a task must have a world model. This is in contrast, for example, to Vafa et al. (2024), Bruce et al. (2024) or LeCun (2024). We draw this distinction for two reasons. First, it seems intuitive: one could easily write a program to perfectly predict next board states of tic-tac-toe based on a move sequence lookup table—yet it seems unhelpful say this system has a "model" of the game. Second, the many heated debates in this area really do revolve around *how* systems work, rather than their accuracy. In fact, often the crucial question isn't whether systems perform well on given inputs, but whether high performance has any chance of generalizing beyond the current test distribution.

## 2 WHAT DO PEOPLE MEAN BY A "MODEL"?

We now describe some of the existing usage of "world model" and adjacent terms. Our focus is specifically on use of the term in the context of internal representations, rather than tests of behavior. The purpose of this section is to underscore the diversity of meanings, and to pick out which themes a minimal definition should capture.

### 2.1 COGNITIVE SCIENCE

#### 2.1.1 MENTAL MODELS

The concept of "mental model" in the background of psychology has an obvious connection to the "world model" we want to define here. Kenneth Craik suggests that the human mind constructs "small-scale models" of reality that it uses to anticipate events (Craik, 1967). Experimentally, O'Keefe & Dostrovsky (1971); Hafting et al. (2005) find that certain neurons, place cells from hippocampus

and grid cells from entorhinal cortex respectively, implement such "world models" of the external world, crucial for the navigation capabilities of rats. A common theme in all of these definitions is the presence of a function from neural activity to some aspect of the real world.

### 2.1.2 REGULATORS

The work of Conant & Ashby (1970) investigates a system that "regulates" some aspect of the world. They prove a theorem which they interpret as saying that a minimal "regulator" will necessarily be an isomorphic model of the world it regulates. The spirit of this paper has been inspirational for many, and the idea of defining a model in terms of an isomorphism is an important point of reference for our definition. On the other hand, their theorem does not seem immediately useful or relevant to the case of neural networks—for one thing, it relies crucially on premises that are highly unlikely to hold in real-world settings.

### 2.1.3 PROBABILISTIC CAUSAL MODELS

An important theme in the cognitive science literature is the idea of a system using or learning probabilistic causal models (among many others, see Lake et al. (2017); Tenenbaum et al. (2011); Friston et al. (2021); Wong et al. (2023)). This is a very strong meaning for a world model. We would like any definition we create to apply to this case, but we believe that even much weaker senses of the word "model" are interesting enough that they deserve to be included as well.

## 2.2 REINFORCEMENT LEARNING

In model-based reinforcement learning (RL), world models, either provided by system designers (Silver et al., 2017) or separately learned (Ha & Schmidhuber, 2018), serve as the foundation for planning. This type of "model" is meant to capture both the state of the environment, and the effect of actions on that state. See Xie et al. (2024) for an excellent overview of the RL perspective.[1]. Model-based RL algorithms have been suggested as a path towards building autonomous intelligent agents such as JEPA (LeCun, 2022). The key here is that the model is either explicitly given, or learned by an explicitly specified module. That is, the system designers determine where the model is, and how it has been learned—there is no mystery involved.[2]

In contrast, model-free algorithms, which launched the field of deep reinforcement learning (Mnih et al., 2013; Schulman et al., 2017), do not have an explicit world model. However, we do see reports of model-free RL developing what look like internal representations of the world. For example, Wijmans et al. (2023) report evidence of neural "maps" in navigation agents. One can view such maps as closely related to animal mental models, and very much in the spirit of the definition we propose in the next section. Note that such representations are close in spirit to the pure state-space models which are the focus of this paper.

## 2.3 SEQUENCE MODELS

The term "world model" appears in several papers on sequence prediction. For example, in Li et al. (2022), a world model is defined as being simple, interpretable, and controllable; the central example of the paper is a relatively concise mapping from internal neural network activations to states of an Othello game board. Here again, we see the themes of a mapping from internal state into world state, as well as the importance of the simplicity of such a map. Similar ideas around state-tracking can be seen in Toshniwal et al. (2022); Li et al. (2021); Patel & Pavlick (2022). These ideas also appear in research that moves beyond games and toy systems, e.g. to representations of perceptual spaces (Abdou et al., 2021). Some researchers have looked for internal models of real geography, as in Vafa et al. (2024), or even of details of the entire geographic world (Gurnee & Tegmark, 2023). The search for internal world-state representations clearly has drawn great interest.

---

[1] As Mark Riedl says (BlueSky, 2024): "I'm an RL guy. To me, a world model maps (state, action) -> state' "

[2] Note that the phrase "world model" is overloaded. We can also examine the "world models" proposed in Ha & Schmidhuber (2018); Bruce et al. (2024) as the neural network $f$ in our setting, and they do not necessarily contain a world model $M$ under our definition.

### 2.3.1 PROBING METHODS

A classic way of investigating internal representations, used in many of the papers above, is **probing**. The idea behind a probe is to train a simple function (typically linear, although sometimes a neural with one or two hidden layers) on a networks internal activations, to see if it can learn to predict some human-interpretable aspect of the environment or data (Alain & Bengio, 2016; Belinkov, 2022).

The probing methodology is a kind of template for the definition we wish to create. It includes, as a first-class citizen, a map from the internal activations of a network to some object of interest. By restricting this map to a predefined "simple" set of functions it avoids trivialities. Furthermore, probing methodology typically includes the use of "interventions" (Belinkov, 2022) to test whether internal representations play a causal role in a network's output. This is an important validity check. Our definition borrows heavily from the probing literature, while adding a few new ideas. For example, the probing literature doesn't always make a clear distinction between "world" and "data."

## 3 DEFINING WORLD MODELS

In this section we provide a careful definition of a world model, which we believe can include most of current usage. Because the final result is complicated, we build up to it in multiple steps.

### 3.1 THE SETTING

Our basic set-up is a neural network represented as a function $f : X \to Y$. Here $X$ and $Y$ are real vector spaces, corresponding to input data and output values, respectively. The input data is the result of observations of a **world** $W$. The world can be any type of object on which we can define a function. The data, $X$, is produced by an **observation function** applied to the world, $\alpha$. In diagrammatic form:

$$
\begin{array}{c}
X \xrightarrow{\ f\ } Y \\
\alpha \uparrow \\
W
\end{array}
$$

To give a concrete example of this setting, consider a familiar MNIST digit classification network. Here $X = \mathbb{R}^{784}$ (the space of 28x28 grayscale images), $f$ is a classifying function learned by a neural network, and $Y = \mathbb{R}^{10}$ is the space of confidences for each digit. The world $W$ is the real world that produced the MNIST drawings—this is necessarily very hard to describe mathematically, involving the process of multiple people drawing digits with pen and paper. Similarly, the observation function $\alpha$ is also hard to describe and very complicated, representing cameras, digitization, etc. Importantly for our argument, we don't need to have a precise mathematical model of $W$ or $\alpha$. In this case that's probably impossible. However, we often do have partial information about $W$. For example, in this case we know that individual people were trying to draw one of ten specific figures.

Of course, we might also be interested in purely mathematical examples. In these cases, we can make all the terms precise. For example, in studying neural networks that are trained on modular addition, the world $W$ might be ordered pairs of numbers mod $n$, and the observation function $\alpha$ might encode the numbers and their mod-$n$ sum into a vector format.

To think of our complex world in a precise way, we can additionally take on the assumption that our physical world is Turing-simulatable. In this case, the world $W$ represents the whole entity of the Turing machine, and an observation function $\alpha$ takes the current tape state as input, computes its observation of interest $X$, and write to an separate paper tape.

### 3.1.1 FACTORING THE FUNCTION $f$

Because we're interested in internal representations, we will "factor" the function via an intermediate vector space, $Z$. In other words, we set $f = f_2 \circ f_1$, where $f_1 : X \to Z$ and $f_2 : Z \to Y$. In diagrammatic form we have:

$$X \xrightarrow{f_1} Z \xrightarrow{f_2} Y$$
$$\alpha \uparrow$$
$$W$$

Smoothing over some details, this is consistent with a wide array of practical neural network architectures. A detailed discussion can be found in Appendix A.

## 3.2 DEFINING "WORLD MODEL"

Within this setting, we can now define what it means for $Z$ to contain a **world model**, $M$. Informally, $M$ needs to simultaneously have a meaningful relation to $Z$ (which contains it) and $W$ (since it models the world). To express this formally, we define a world model to be a vector space $M$, along with two functions $\varphi_1 : W \to M$ and $g : Z \to M$ such that $\varphi_1 = g \circ f_1 \circ \alpha$ . (The subscript of $\varphi_1$ is meant to parallel $f_1$, and will come in handy later.) Similarly, $g$ is a projection from $Z$ to $M$. Again, this is easiest to see as a commutative diagram[3]:

$$X \xrightarrow{f_1} Z \xrightarrow{f_2} Y$$
$$\alpha \uparrow \qquad \downarrow g$$
$$W \xrightarrow{\varphi_1} M$$

Intuitively, we can think of $\varphi_1$ as a kind of "modeling" function that maps the full world $W$ onto a (hopefully easier to understand) image $M$. However, we must add some restrictions on this function to make the definition non-vacuous.

### 3.2.1 RESTRICTED CLASSES OF FUNCTIONS

To make the definition of world model interesting we need to put constraints on $\varphi_1$ and $g$. One issue is that we could take $g$ to be any function at all on $Z$, and then simply define $\varphi$ to be $g \circ f_1 \circ \alpha$, and $M$ as the image of $\varphi$. That hardly seems like an interesting example of a world model! A second issue is that we'd like our definition to say something about nontrivial about $f$. However, if the function $g$ is itself extremely complex, it's not clear what we have learned about $f$ from the existence of a world model[4].

To avoid this kind of triviality, we stipulate that both $\varphi_1$ and $g$ must belong to pre-specified classes of "simple" functions, $\mathcal{F}_W$ and $\mathcal{F}_Z$. For instance, a common case is that the function $g$ is constrained to be linear—this corresponds to the "linear probing" technique. Other classes of functions include projection onto a single coordinate (i.e., looking at the value of a single neuron) or two-layer MLPs. Defining what it means for $\varphi_1$, a function whose domain is $W$, to be "simple" is potentially harder. If the world $W$ is mathematically precise (as in modular addition) then we can fall back on standard classes of mathematically simple functions.

However, in most applications of interest we may not know the mathematical form of $W$. In this case one option is to consider the class of functions that are **human-interpretable**—that is, describable in terms a person can reasonably understand. The catch, of course, is that this family is somewhat vague (we have no exhaustive description of its members) and many human-interpretable terms, like "is grammatical" or "is socially toxic," are hard to formalize. In practice we can often operationalize this type of function via human labelers, but it's important to be clear that in this part of the diagram we may need to leave the rigorous world of pure mathematics.

---

[3]If you haven't seen this kind of diagram before, the idea is that moving from one space to another along an arrow means you're applying the function labeling the arrow. If there are two paths to move from one space to another, the implication is those two different function compositions are equal.

[4]Of course, we would be able to deduce that $X$ has enough information to reconstruct $M$, and that $f$ at least did not destroy this information. But this is largely a statement about the data $X$, and it doesn't imply that $f$ has done any sort of interesting computation.

### 3.3 A NOTE ON APPROXIMATION AND MIXED BEHAVIOR

The definitions above use equations. Yet it's a rare neural network that computes its target task perfectly. As a result, using strict equalities in the definition of a world model isn't likely to be productive. It's more realistic to look for approximations rather than exact equality. For simplicity, we continue to write definitions in terms of equations. However, it should be understood that the definitions still apply to a system in which equality holds to a good approximation.

What constitutes a "good approximation" will of course depend on context. In fact, there's a wide range of views in the literature. One investigation into internal models of an LLM Li et al. (2021) suggests that a world model exists based on functions that do significantly better than chance, but are far from perfectly accurate. That actually seems reasonable, as sufficient evidence that a network is doing more than just memorizing superficial statistics.

On the other hand, it leaves open the idea that the network is using some combination of a world model, and miscellaneous heuristics. Indeed, it seems possible that many neural networks do exactly that. For instance, careful investigations into the Othello model introduced by Li et al. (2022) indicate that despite the existence of an internal world model, the network uses a variety of brittle heuristics in its predictions (Nanda et al., 2023b; Chiu et al., 2023). In summary, if a neural net meets our definition with approximations instead of equations, that should be seen evidence for a world model that only partially explains the system's behavior.

### 3.4 AN EXAMPLE: A SENTIMENT MODEL

A classic example of these ideas in action comes from Radford et al. (2017), which describes a neural network trained to predict the next character of Amazon reviews [5] The authors found a single neuron in this system that was a state-of-the-art sentiment predictor [6]. We may view this neuron's activation level as a model of the world, specifically a model of the writer's state of mind. The following table shows how our formalism can describe this result.

| Symbol | Full world model definitions for the case of the sentiment neuron |
|---|---|
| $W$ | World where reviews were written and collected, including writers and their emotions |
| $\alpha$ | Process of writing and collecting reviews |
| $X$ | Review text |
| $Y$ | Next character in text |
| $f_1$ | Intermediate calculations of LSTM |
| $Z$ | Activations of neurons |
| $f_2$ | Final calculations of LSTM |
| $M$ | Sentiment of writer |
| $\mathcal{F}_W$ | Human-computable functions |
| $\varphi_1$ | Human labeling of sentiment |
| $\mathcal{F}_Z$ | Projections onto coordinates |
| $g$ | Projection onto coordinate representing "sentiment neuron" |

Table 1: A sentiment neuron as a world model, based on neural network described in Radford et al. (2017)

### 3.5 AVOIDING TRIVIALITIES

Although the definition in the previous section covers many cases of interest, it also includes a number of trivial cases. To rule out these trivialities, we describe a few additional conditions. Conveniently, this helps us make precise several other frequently used terms.

---

[5]Compare to Andreas (2022), which has a detailed analysis of how this model may relate to communicative intent.

[6]One paper has raised questions about the role of the sentiment neuron in prediction (Donnelly & Roegiest, 2019); however, the results are unclear and don't affect the thrust of our argument here.

### 3.5.1 "LEARNED" WORLD MODELS

The reason we investigate world models is our interest in the representation $Z$. However, if the data in $X$ itself effectively contains a model of $W$, then the fact that $Z$ contains one too would be of little consequence. To capture this idea, we start by considering a restricted set $\mathcal{F}_X$ of functions $X \to M$ that parallels $\mathcal{F}_Z$. For example, if $\mathcal{F}_Z$ consists of linear functions on $Z$, then $\mathcal{F}_X$ might be linear functions on $X$. Then we say a world model is **learned** if there does not exist a function $h \in \mathcal{F}_X$, such that $h = g \circ f_1$. In diagrammatic form we may put it like this:

$$
\begin{array}{ccccc}
X & \xrightarrow{\ f_1\ } & Z & \xrightarrow{\ f_2\ } & Y \\
\alpha \uparrow & \searrow & \downarrow g & & \\
W & \xrightarrow{\ \varphi_1\ } & M & &
\end{array}
$$

This nontriviality condition is surprisingly important, because in practice it's easy for the data in $X$ to have a linear projection to a seemingly sophisticated world model. We'll give two examples to show why the "learned model" distinction is stronger than it may initially seem.

**Example 1. Word co-occurrence statistics can model the world.** Word embeddings, such as word2vec (Mikolov et al., 2013) or Glove (Pennington et al., 2014), famously can be used to perform word analogy tasks using vector arithmetic. For example, a particular direction in space might correspond to a gender distinction, or to a country-capital relationship. This might seem to be the result of deeply complicated functions (neural network or other) transforming basic co-occurrence statistics into a linear model of key aspects of text.

However, co-occurrence statistics already have a rough linear model of these concepts! This has been observed at least since the invention of latent semantic analysis (Dumais, 2004). Indeed, the report on Glove (Pennington et al., 2014) provides SVD-based linear projections of co-occurrence statistics as baselines, which are remarkably close to the final model they use[7]. The implication is that if the data $X$ for a neural network consists of something as seemingly basic as co-occurrence vectors, then it already contains a linear model of aspects of the world ranging from social structures to geography. Any intermediate representations of these properties may not be truly learned—they were in the data from the beginning. For example, Gurnee & Tegmark (2023) find that the intermediate representations of language models contain linear representations of geographic and chronological features of the world. More work would be needed, however, to deduce that this is *learned* world model.

**Example 2. Partial views of a dynamical system often model the full system.** Consider a "world" consisting of a discrete dynamical system defined by a smooth map $F : \mathbb{R}^n \to \mathbb{R}^n$. Following Takens (1980), suppose that our input space $X$ consists of one-dimensional "observations" of an orbit of a point $x$ under the action of $F$. That is, the input is a sequence $(\alpha(x), \alpha(F(x)), \ldots, \alpha(F^k(x)))$, where $\alpha : \mathbb{R}^n \to \mathbb{R}$ is an "observation function" that satisfies a few mild non-degeneracy conditions. Each such sequence clearly loses a huge amount of information from the original dynamical trajectory, since $\alpha$ produces just a single real value.

Now imagine training a neural network to predict the next observation in this sequence, and then discovering that intermediate representations of the input points form a diffeomorphic image of the dynamics in the original space $\mathbb{R}^n$. At first this might seem like an incredible feat of learning! However, as long as the sequences are reasonably long compared to the dimension $n$, then such a diffeomorphic image is likely already present in the data, simply by concatenating windows of $2n + 1$ observations. This is the content of Takens's famous reconstruction theorem.

**Comparability of $\mathcal{F}_X$ and $\mathcal{F}_Z$.** There are some subtle issues that might arise in defining and comparing "simple" functions on to different spaces . If the two spaces have vastly different dimensionalities, for example, then the spaces of linear functions will have vastly different numbers of parameters. If this difference becomes an important concern, one way to make a direct comparison

---

[7]Both of these examples operate only a low-dimensional subspace of highest variance in word co-occurrence vectors. That's an important fact, since the original co-occurrence vectors are extremely high-dimensional and unlikely to have linear dependencies. Thus for trivial reasons there is almost certainly a linear projection from the full high-dimensional space to any model of moderate dimensionality. We emphasize that the linear structure seen in the quoted papers is more interesting than that.

is to consider a random "control function" $f_{\text{random}} : X \to Z$, and test whether there is a function $g' \in \mathcal{F}_Z$ such that $g' \circ f_{\text{random}} = g \circ f_1$. If such a $g'$ can't be learned, that strongly suggests that $f_1$ is performing a nontrivial transformation on $X$.

### 3.5.2 "EMERGENT" WORLD MODELS

A second uninteresting situation arises if the output $Y$ contains something like a world model. In this case it's unsurprising if $Z$ models $W$—it's almost forced by the task. For instance, if the task is to measure sentiment from text, then it's uninteresting to know that an intermediate representation has information about sentiment.

In other words, we are most interested in world models $M$ where **there does not exist** a function $h : Y \to M$, in a restricted class $\mathcal{F}_Y$, such that $h \circ f_2 = g$. (Again, $\mathcal{F}_Y$ should parallel the definition of $\mathcal{F}_Z$) To indicate that learning a world model isn't part of the target task, we call it an **emergent world model**. In diagrammatic form:

$$
\begin{array}{ccc}
X & \xrightarrow{\ f_1\ } Z \xrightarrow{\ f_2\ } & Y \\
\alpha \uparrow & \quad \downarrow g \ \diagdown \!\!\!\!\!\diagup & \\
W & \xrightarrow{\ \varphi_1\ } M &
\end{array}
$$

Aside from avoiding trivialities, one reason we give this definition is that the word "emergent" has been used in many different ways. We wish to distinguish our usage especially from the idea that an emergent property being one that appears "suddenly" during training (Schaeffer et al., 2023). Instead, we wish to emphasize that the world model isn't reducible to a trivial consequence of the form of either the input or the output. This is in line with philosophical definitions of the term (O'Connor, 2021).

## 4 CAUSAL WORLD MODELS

Suppose we're lucky enough to identify a learned emergent world model in a system's internal representation $Z$. How do we know that this model actually relates to the function $f_2$? It's certainly possible there's no relation at all. In fact, the only stipulation we've made so far that connects the model $M$ and the output space $Y$ is negative: that for an emergent model there is *no* simple function taking $Y$ to $M$.

From a theoretical perspective, it's easy to imagine functions where knowing about a world model gives no information about how a function is computed. For example, suppose that $Z = Z_1 \oplus Z_2$, where $Z_1$ contains a world model, but $f_2$ only looks at data in $Z_2$. It's also easy to see how this could be a problem in practice. Even a seemingly "small" class of functions, such as linear maps from $Z$ to $M$, is fairly powerful when the dimensions of $Z$ and $M$ are high enough. It's not a stretch to imagine that there will be *some* linear map can pick out *some* simple aspect of the world $W$, even though that aspect is never used by $f_2$.

We're concerned with causality for two reasons. Scientifically, spurious representations aren't that interesting, since it's not clear how they help us understand what our system is doing. Just as important is the engineering perspective: if we want to use our understanding to control a neural network, then causality clearly matters. How can we capture the idea that a world model matters? Below we provide two definitions for "complete" and "partial" causality of a world model.

### 4.1 COMPLETE CAUSAL WORLD MODELS

A major interpretability goal is to find a simple intermediate representation that completely determines the output of a neural network. One way to express this formally is to require the existence of a function $\varphi_2 : M \to Y$ such that $\varphi_2 \circ g = f_2$. Or, in diagrammatic form:

$$
\begin{array}{ccc}
X & \xrightarrow{\ f_1\ } Z \xrightarrow{\ f_2\ } & Y \\
\alpha \uparrow & \quad \downarrow g \ \diagup \varphi_2 & \\
W & \xrightarrow{\ \varphi_1\ } M &
\end{array}
$$

If there is such an $\varphi_2$ then we say the model $M$ is **causally complete**, since its value completely determines the output of the model. Assuming that the world model is much simpler than the world itself, it gives us two valuable things. First, we can fully characterize the function $f_2 \circ f_1$ in natural terms: as a function operating on $M$, rather than $W$ or its noisy proxy $X$. Second, it provides a means of control: if we modify an internal representation $r \in Z$, then we can predict the effect of that modification in terms of $M$.

Completeness in this sense is a strong condition. Given the complexity of the real world, it's unlikely we'll ever find, say, a complete world model inside of ChatGPT. However, for simple synthetic tasks this may be possible. One example is the type of modular addition network considered in Nanda et al. (2023a).

### 4.2 Causal world models

In practice, completeness may be too high an ideal. A weaker condition, however, can still capture the ideal of a world model that has a nontrivial causal effect on a computation. Let's return to the example of the "sentiment neuron" in Radford et al. (2017). As discussed, the researchers on that project found a neuron whose activation reflected the sentiment of the text it was given. But they didn't stop there. They did one more critical experiment: during inference, they performed an intervention, fixing the value of this neuron at a value of either zero or one. When fixed at zero, the system generated negative-sentiment completions (as measured by human labelers). Fixing the neuron at a value of one produced positive completions.

We may view the sentiment neuron as a model of the world (the reviewer's overall emotion) which has a causal effect on one aspect of the output (the human-judged sentiment). However, we don't have a truly complete model, since the sentiment neuron alone generally doesn't determine the next character.

We can formalize this situation by defining a simplified aspect of the output $A$ (here, its sentiment) and a function $h : Y \to A$ (here, human judgment of output sentiment). The finding that the representation is causal is equivalent to the fact that $h \circ f_2 = \varphi_2 \circ g$. In diagram form:

$$
\begin{array}{ccccc}
X & \xrightarrow{f_1} & Z & \xrightarrow{f_2} & Y \\
\alpha \uparrow & & \downarrow g & & \downarrow h \\
W & \xrightarrow{\varphi_1} & M & \xrightarrow{\varphi_2} & A
\end{array}
$$

This diagram represents what we call a (potentially incomplete) **causal world model**. To avoid trivialities, we stipulate that $h \circ f_2 \circ f_1$ is not constant. That is, it needs to capture some varying aspect of the output $Y$, for which the model $M$ is predictive. One interpretation of this definition is that, with a world model identified, we can move from thinking about the top row (the actual data and neural network) to the bottom row (the higher-level of picking out a simplified model of the world, and operating on that.)

### 4.3 Causality and the transformer architecture

Transformer architectures can add one more layer of complexity to the picture we have sketched so far. First, the residual stream for a given token is often viewed as a shared resource across layers (Elhage et al., 2021). It's plausible that any world model it contains is read from, and written to, at multiple layers. In this case, even when it is possible to read a world model from just one layer (consistent with the framework we've laid out), it is possible that an effective intervention on the world model may require making changes as multiple layers. See subsection A.1 for a full description of this phenomenon.

## 5 Local world models

Today's machine learning systems are often trained on multiple tasks, complicating the notion of a single world model. Our last definition is meant to express the idea that a world model may be highly contextual, but still useful. As a motivating example, consider Li et al. (2021), which found what

they called "implicit representations of meaning" in a tuned language model. In particular, when the model was given English descriptions of a series of actions in an specific constrained domain, they found evidence for internal representations of the state of this world as a result of these actions. In the notation we've developed, the underlying state is $M$ and answers about the state lie in $A$. The generated text is $Y$, and $h$ is a function (performed in this case by humans) translating the text into the modeled state.

This is close to the world model described in the previous section, but with one caveat: the system used by Li et al. (2021) was a general language model, and there's no evidence that the world model $M$ had any explanatory power when given text input that did not relate to the domain it was tuned for. In other words, the world model may be operative only for a subset of world states, $W'$. We call this a **local world model**, and represent it in diagrammatic form as follows:

$$
\begin{array}{ccccc}
X & \xrightarrow{\;f_1\;} & Z & \xrightarrow{\;f_2\;} & Y \\
{\scriptstyle\alpha}\big\uparrow & & \big\downarrow{\scriptstyle g} & & \big\downarrow{\scriptstyle h} \\
W' \subset W & \xrightarrow{\;\varphi_1\;} & M & \xrightarrow{\;\varphi_2\;} & A
\end{array}
$$

For general-purpose systems, we might speculate that this type of contextual model is the most we can hope to find. And more broadly, we may wish to restrict consideration of any of the functions in our commutative diagram to a specific subspace or subset of data. For example, it will often be the case that the image of $f_1$ is not the whole of $Z$. To establish causality, it makes sense to test if $\varphi_2 \circ g = h \circ f_2$ outside of the set $f_1(X)$. However, it may be enough to test only points in a local neighborhood of this set; requiring equality for points that fall far out of distribution may be an unnecessary restriction.

## 6 CONCLUSION

We have presented a definition of the term "world model," which we hope can unify and clarify a broad set of interpretability research. The mathematical framework of the definition means that it is straightforward to operationalize experimentally. The general template for the definition comes from the probing literature, which is already familiar to researchers in the field.

There are several ways in which it might be useful to extend this definition. Currently we've aimed for a "minimal interesting" definition. As a result, we've proposed a relatively weak set of criteria. An important extension would be to find analogous definitions of what it means to learn a joint model of states, actions, and the results of actions.

To summarize, our hope is that this definition can clarify some of the controversy around what it is that LLMs and similarly powerful neural networks are really learning. Asking whether a network "understands" its input, for instance, is a recipe for (ironically) misunderstandings. Asking instead whether it builds a world model, as described here, leads to questions that can be resolved scientifically.

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

# APPENDIX

## A  ARCHITECTURAL DETAILS OF DECOMPOSING NEURAL NETWORKS

Our definition is based on a factorization of a neural network of the form $X \rightarrow Z \rightarrow Y$. For a standard feedforward network, it's clear how to do this: we can just take $Z$ to be the output of an intermediate layer, which is at the same time the input to the subsequent layer. For any network with residual connections, it is important to take out $Z$ after the addition operation to form a cut-off.

For sequence processing models like RNN and SSM (Hochreiter & Schmidhuber, 1997; Vaswani et al., 2017; Gu & Dao, 2023), representations of all input tokens are constructed recursively and serve naturally as $Z$. For transformer (Vaswani et al., 2017), we can take the mid-layer embeddings of all input tokens at an arbitrary layer as $Z$, which effectively cut off the computation graph from input to output. As a special case, if the last layer is chosen, only the last-token embedding is needed to form $Z$. Some probing techniques in the literature probe only at the embeddings of specific tokens, e.g. last tokens, which still falls into the framework since we can take the probing function $g$ to first discard all features in $Z$ except that for the last time step.

In general, the factoring of a function $f$ into $f_1$ and $f_2$ by a cut-off representation $Z$ discussed in subsubsection 3.1.1 is a generic approach for examining a wide range of neural networks. This formulation is closely related to existing ways of understanding such networks.

### A.1  INTERVENTION AT MULTIPLE LAYERS: THE CASE OF OTHELLO-GPT

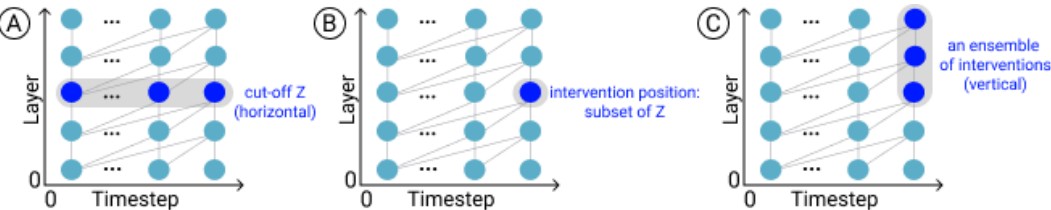

Figure 2: (A) For a transformer, the natural cut-off point is the entirety of the features of all tokens at any given layer. (B) However, in the intervention process in Li et al. (2022), a specific subset of the cut-off $Z$ is targeted. Nevertheless, it's imaginable that a novel intervention technique could be devised to operate on the whole cut-off $Z$ to achieve better intervention effects. (C) Differently, Li et al. (2022) took an alternative path and carried out an ensemble of interventions across multiple cut-off layers, in each of which only the feature of the last token is edited.

One of the applications of such world models is that we can deliberately modify $Z$ during the computation process so that the conceived world model $M$ is modified correspondingly, thus the prediction $Y$ is changed accordingly (Li et al., 2021; 2022). As an example, in Othello-GPT (Li et al., 2022), the board states could be probed at the features of the last token across layers with varying accuracies; and this board state representation is editable—the authors were able to purposefully change the features for the last token across several layers to reflect a counterfactual board state so that the prediction of the next move is altered in a predictable way.

However, this approach seemingly runs against the definition of world model in the main prose. Ideally the intervention should happen a cut-off $Z$, which is horizontal in transformer (A in Figure 2); but in Li et al. (2022), the intervention is done vertically (C in Figure 2), where the intervened features do not form a cut-off set of the forwarding computation of the transformer. Below, we will show that this conflict is only apparent at first glance.

Let's first consider editing a single layer (B in Figure 2). In this case, the intervention still falls within our framework since it is at the freedom of the intervention to operate on all or only a subset of $Z$. In reality, the intervention only happens on a subset (the last token) of $Z$ as an engineering choice made by Li et al. (2022). As seen in the Figure 3 of Othello-GPT paper, intervening only on one layer still yielded a positive editing effect, making a conceptual sense. However, since the features of the previous time steps are not edited, leaving much unedited (factual, compared to counterfactual

information in the edited token) information to be leaked into later layers, the intervention success rate is not as high as editing across layers. It is also likely that there is a to-be-invented novel intervention technique that edits horizontally on a single layer and yields strong effect.

Now, how do we reconcile the fact that the intervention is repeated across layers? We could imagine an ensemble of choke points at different layers, e.g. $X \to Z_1 \to Y$ at the first layer and $X \to Z_2 \to Y$ at the second layer. The intervention scheme (C in Figure 2) is then an ensemble of interventions on different layers. The fact that this ensemble strategy provides an improved intervention effectiveness corroborates the probing results that the world model exists at each layer-wise cut-off $Z$, up to varying accuracies.

In a way, we can think of the residual stream of a transformer as the scratch paper (or paper tape) of the model itself. On this scratch paper, a world model is synthesized gradually by multiple layers and at the same time used by multiple layers to populate other details of the world model or to make prediction. Defining $Z$ to be a single layer is a useful way of thinking about *reading* a world model, but it because models may be built and used across multiple layers, it we may need to intervene at more than one layer to *control* a world model.

