# OpenReview forum: "What Does it Mean for a Neural Network  to Learn a "World Model"?"
_ICLR.cc/2025/Conference — ICLR 2025 Conference Withdrawn Submission_

### Official Review · Reviewer_aeim · 2024-11-02

**Soundness:** 2
**Presentation:** 3
**Contribution:** 2
**Rating:** 5
**Confidence:** 2

**Summary:**

The paper proposes an abstract criterion for claiming that a world models is being implemented in a neural network. The definition is based on the existence of some commutative functions mapping the hidden representation to a salient aspect of the network's behavior. Models should be 'interpretable' and 'causal'; interpretable meaning that the model is some pre-specified type of function of the world/representation, and causal meaning that the model predicts behavior just as the representation predicts outputs.

**Strengths:**

The paper does an extensive review of the literature, and motivates each aspect of the central diagram.

**Weaknesses:**

Usually for a definition to become accepted, it should demonstrate that it includes most clear-cut cases and excludes others. Some of this is done for intermediate steps (like the sentiment neuron, and word embeddings), but it feels important to do this for the full definition. That is, what kinds of models/analyses does the diagram in Figure 1 accept and reject?

As it stands, the definition remains very abstract, and has not demonstrated its utility with theoretical results or for empirical analysis. While the authors say it is 'straightforward to operationalize experimentally', that isn't obvious to me given its level of abstraction -- the hard part seems to be precisely in applying the high-level concept to specific cases. In other words, its not clear to me how much a commutative diagram by itself, without specifying what the sets or functions are, allows us to judge networks theoretically or develop/understand methods for interpretability. If there is a good demonstration of this, I think it should be included and highlighted.

**Questions:**

Is the idea behind this definition that, for any analysis trying to infer a world model, one should be able to point to components of the analysis and say, this is $\phi_1$, this is $W$, this is $g$, etc.? Or, maybe equivalently, that $Z$ only represents a world model if a commutative diagram like that exists?

---

> ### Author Response · Authors · 2024-11-14
> **Response to reviewer aeim**
>
> Thank you for your helpful comments. Your final question is exactly on target: we hope that future researchers who investigate world models can say explicitly which functions they are studying. Our sense is that many papers already do this, but use inconsistent terminology that makes it hard to compare results. Other papers do not do this, but use similar terminology, which is even more confusing. We described a few such cases in the general comments, and can put additional examples in the paper.

---

### Official Review · Reviewer_CbCD · 2024-11-04

**Soundness:** 2
**Presentation:** 3
**Contribution:** 2
**Rating:** 3
**Confidence:** 3

**Summary:**

The authors propose a set of criteria for saying a neural net learns a "world model".  This is a purely theoretical paper with no experiments.

**Strengths:**

It is helpful to have clear definitions of what we mean by a "world model".

**Weaknesses:**

Since I am more on the experiment side, I don't understand what follows from this paper.  How does the proposal relate to prior work?  Is the proposed framework falsifiable?  What follows from this contribution?

The word "understanding" is mentioned nearly a dozen times, but I don't know what that means.  Maybe that is the point of the last paragraph.  But that begs the question, what is a world model?  Is that what you propose?  But that sounds circular.

The first paragraph of the conclusion claims to have unified work on interpretability.  The paper mentions interpretability nearly a dozen times but I'm having trouble seeing the connection between this paper and those references.

**Questions:**

Can you help me understand the consequences of this paper?  What follows from this contribution?

Would it be possible to test your ideas?  If so, how?  What would an experiment look like?

---

> ### Author Response · Authors · 2024-11-14
> **Response to reviewer CbCD**
>
> Thank you for pointing out some important gaps. We’ll connect the sections to interpretability work more clearly: although we’ve tried to give key examples (the sentiment neuron, etc.) we can certainly provide more. We’ve tried to address your concerns in the general comments above, but please let us know if there are additional issues you see.

---

> > ### Comment · Reviewer_CbCD · 2024-11-26
> > **nothing more to add**
> >
> > I think we have discussed this sufficiently.  The system is complaining that I am not commenting enough so I am adding this stupid non-comment.

---

### Official Review · Reviewer_XPCw · 2024-11-04

**Soundness:** 4
**Presentation:** 4
**Contribution:** 3
**Rating:** 8
**Confidence:** 4

**Summary:**

This work seeks to establish minimum criteria by which one can say that a neural network encodes a world model. By establishing these criteria, this work aims to provide common  (and precise) language by which interpretability researchers can discuss the inner mechanisms of neural networks.

**Strengths:**

The goal of this work is necessary and important for shaping the discourse around neural networks, model interpretability, and the relationship between the field of deep learning and other fields that require more high-level/abstract models (such as cognitive science). Importantly, this fairly high-level paper is grounded in (and makes reference to) recent empirical work in interpretability. The paper is extremely well-written, clear-eyed and explicit about its purview and limitations, and ultimately succeeds in its goals.

**Weaknesses:**

The authors refer to the debate concerning whether LLMs actually understand text (523), and also mention that world models might connect to this issue (055). Making this connection a bit more explicit, even in an appendix, would be enlightening. This would potentially provide an important starting point for reframing the debate around understanding in more scientific terms.

**Questions:**

Why do the authors say that learning probabilistic causal models is “a very strong meaning for a world model” (125)? In general, this section might be expanded a bit.

---

> ### Author Response · Authors · 2024-11-14
> **Response to XPCw**
>
> Thank you for these useful comments! As described in the general comment above, we can definitely expand on how “world models” relate to questions of “understanding.” To answer your question about probabilistic causal models: our wording here is certainly confusing. Essentially, we’re trying to be cautious in our definition, and to be clear that we’re proposing a weaker type of world model (e.g., one would not necessarily be able to answer arbitrary counterfactual questions via the world models we describe.)

---

### Official Review · Reviewer_Y1Mh · 2024-11-04

**Soundness:** 2
**Presentation:** 1
**Contribution:** 2
**Rating:** 3
**Confidence:** 4

**Summary:**

The paper provides a conceptual framework for defining latent world models potentially learned by neural networks. This framework defines world models by their position in a commutative diagram relating to the world being implicitly modeled, the data (sampled from this world) that the network is trained on, and the latent representations learned by the network. The paper also includes several guidelines for instantiating this framework, like how to avoid trivial instantiations or study "local" world models covering narrower contexts than the network was trained on.

**Strengths:**

The paper clearly motivates and describes the underlying problem: how should we conceptualize the notion of "world models" in the context of latent neural network representations? This problem is indeed important and deserving of study, and the core idea of defining a commutative diagram over a network's training data, the "world" that training data is sampled from, and latent representations learned by the network is innovative.

**Weaknesses:**

Beyond simply defining the research question (how should we conceptualize the notion of "world models" in the context of latent neural network representations?) and drawing a commutative diagram that is useful for conceptualizing this question, **the paper does not provide a clear or meaningful contribution.**
While articulating the central question is helpful, this alone is not a sufficient contribution for an ICLR conference paper.

Indeed, the paper is styled more as a blog post than a conference paper: beyond its highly informal prose and prominent citations of tweets (Lecun, 2024) and blog posts (Andreas, 2024), it also **lacks the theoretical rigor that is necessary for a conceptual submission such as this one**; and it is not obvious that the proposed framework could be adequately formalized at all. For example:
- In sec 3.2.1, the framework requres "simple function classes" $F_W$ and $F_Z$ that restrict the range of functions that map between certain nodes in the commutative diagram in order to avoid a trivial definition. But "simple" is never formally defined -- instead, a few possible examples are provided without justification -- and it is never explained *how* requiring these functions to be "simple" actually resolves the triviality.
    - Furthermore, one of the examples provided in sec 3.2.1 of one such "simple" function class is that of two-layer MLPs, which are universal approximators; or in sec 3.4, another example for $F_W$ is listed as the space of "human computable functions". Either case would seem to strain any reasonable interpretation of "simple".
- Section 3.5.1 is concerned with preventing another possible triviality, this time the existence of another "simple" function that maps directly from inputs to a potential world model; but it is unclear both (1) why this is a problem, and (2) how the provided definition prevents this triviality.
    - E.g., word co-occurrence statistics are provided as an example of a "trivial" world model, as many interesting features of a potential world model can be approximated using linear projections from such statistics. However, this is simply another way of discussing the "distributional hypothesis" underpinning modern LLMs -- i.e., that sufficient knowledge of word co-occurrences is sufficient to understand much of natural-language semantics -- and the paper considers LLMs as serious candidates for learning world models. Why, then, is the existence of "simple functions" mapping from co-occurrence statistics to certain world model features understood as presenting a "trivial" case to be avoided?

Another key weakness is that the paper fails to reference several closely related works and relevant areas of study. For example:
- [1, 2] are also centrally concerned with conceptualizing how world models should be understood in the context of foundation models, and [2] also focuses on the intersection of interpretability and world modeling.
- The description in section 2.1 of "world models" as studied in cognitive science is lacking. For instance, predictive coding is one of the leading formalizations of world models in cognitive science [3,4], but predictive coding is never discussed.
- The "random control function" proposed in lines 372-381 appears to be equivalent to "control probes" as defined by [5], but [5] is never cited. Note that, on line 74-75, it is stated that "much of this paper may be seen as a reframing of ideas in Belinkov (2022)", and Belinkov (2022) discusses [5] at length. Thus, the failure to cite [5] is particularly concerning, and may be a sign of plagiarism.

[1] Bisk, Y., Holtzman, A., Thomason, J., Andreas, J., Bengio, Y., Chai, J., ... & Turian, J. (2020, November). Experience Grounds Language. In Proceedings of the 2020 Conference on Empirical Methods in Natural Language Processing (EMNLP) (pp. 8718-8735).

[2] Ruthenis, T. (2023, January). World-model interpretability is all we need. In AI Alignment Forum.

[3] Millidge, B., Seth, A., & Buckley, C. L. (2021). Predictive coding: a theoretical and experimental review. arXiv preprint arXiv:2107.12979.

[4] Taniguchi, T., Murata, S., Suzuki, M., Ognibene, D., Lanillos, P., Ugur, E., ... & Pezzulo, G. (2023). World models and predictive coding for cognitive and developmental robotics: Frontiers and challenges. Advanced Robotics, 37(13), 780-806.

[5] Hewitt, J., & Liang, P. (2019, November). Designing and Interpreting Probes with Control Tasks. In Proceedings of the 2019 Conference on Empirical Methods in Natural Language Processing and the 9th International Joint Conference on Natural Language Processing (EMNLP-IJCNLP) (pp. 2733-2743).

**Questions:**

Definitions:
- As discussed above, how does constraining $F_W$ and $F_Z$ to be "simple function classes" resolve the triviality discussed in section 3.2.1?
- On line 328-329, what does it mean that "$F_X$ parallels $F_Z$" (328-329)? Is it simply that the two function classes should be equivalent, or satisfy some shared notion of simplicity? As with the topic of "simple function classes" throughout the paper, this should be formalized.
- How should "A" in figure 1 (bottom right side of commutative diagram) be interpreted in any case outside of robotics? I do not see any examples of "A" provided elsewhere, including in Table 1; and it is not clear how, why, or whether "A" and "Y" would be different in any other case. If A and Y are collapsed into the same node, does this lead to other potential problems with the diagram?

Conceptual question:
- The commutative diagram is discussed in terms of absolute equality (e.g., as in line 229-230). Is it possible to relax this assumption and allow for commutation to be approximate instead of exact? E.g., it seems that, especially if the intent is to make extensive use of "simple" functions under this framework, it would be more reasonable to expect approximations than learning exact mapping functions. What would the consequences of such approximation be?

**Details Of Ethics Concerns:**

The "random control function" proposed in lines 372-381 appears to be equivalent to "control probes" as defined by [5], but [5] is never cited. Note that, on line 74-75, it is stated that "much of this paper may be seen as a reframing of ideas in Belinkov (2022)", and Belinkov (2022) discusses [5] at length. Thus, the failure to cite [5] is particularly concerning, and may be a sign of plagiarism.

---

> ### Author Response · Authors · 2024-11-14
> **Response to Reviewer Y1Mh**
>
> We appreciate the close reading of the paper and multiple important conceptual comments. And we are especially grateful for the list of references. We will add discussion of these to the manuscript. In particular, thank you for pointing out the omission of a reference to [5], which we will certainly add. To be clear, we do not intend to plagiarize! On the contrary, we intend for our framework to synthesize and credit existing literature wherever appropriate—hopefully this goal comes through in the spirit of the paper overall.
>
> Section 3.2.1, “simple function classes”: You’re right: our description is confusing. We do want to leave some flexibility in the specific type of function, since it may be domain-dependent. However—and this might be the key point of confusion—the intention was to focus on strongly restrictive classes of functions for g and h. (For example, for a multi-layer MLP, we would assume a reasonable bound on the number of neurons, rather than allowing an arbitrarily large MLP that could be a universal function approximator.) We will make clear in any revisions that we assume a meaningfully restrictive class of “probe” functions, e.g. with a much smaller dimensionality / number of parameters than the underlying network being studied.
>
> Section 3.5.1: Our goal here was to define what it means for a world model to be “learned.” The intuition we want to capture is that this means the neural net must transform the initial data in a nontrivial way. If a world model can be derived from the initial data via a linear function, for example, then one would be hard-pressed to say the network has “learned” a nontrivial representation of the world.
>
> Addressing further questions:
> * Lines 328-329: Good point: the word “parallels” is a poor choice of words. In a revision, we’ll make clear that we mean to aim to consider function classes with roughly the same number of parameters in both cases. The idea is to verify that the function f_1 is doing something nontrivial. (The material in 370-380, where we will reference [5], is meant to handle cases where comparable parameter count may not be easy to achieve.)
> * Thank you! Excellent catch on Table 1: we somehow omitted a key row, which is that A should represent the sentiment of the text (as opposed to the mind of the writer)! This example was actually a major motivation for adding “A” to the framework. In general, we want to capture the idea that a particular world model may only be relevant to one aspect of the output; “A” is meant to represent just that aspect. We will clarify this in any revision. Also note that A may indeed be the same as Y in practice—this is the case described in 4.1, of a “complete causal model”.
> * Conceptual question on use of pure equalities: we agree that this is an important point, and one we had attempted to address in the current section 3.3. Of course we’d welcome ideas on where this section falls short.

---

> > ### Comment · Reviewer_Y1Mh · 2024-11-27
> > **Response to rebuttal**
> >
> > Thank you for your response. I have read your rebuttal and general response, and while I appreciate your clarifications regarding some of my questions, I still believe that the paper does not provide a clear or meaningful contribution.
> >
> > Given revisions to address the key missing citations and unclear definitions discussed in my original review and the authors' response, I believe this paper could be made into an interesting blog post or perspective piece -- e.g., see Frontiers [blog post](https://www.frontiersin.org/news/2016/10/27/blog-guidelines) or [perspective](https://www.frontiersin.org/journals/artificial-intelligence/for-authors/article-types) submission types. But given the lack of substantive contribution, this submission would clearly be inappropriate as an ICLR conference paper.

---

### Author Response · Authors · 2024-11-14
**General response to reviewer comments**

Thank you to all reviewers for helpful comments! We’ll answer some general questions first, and then address some specifics of each reviewer’s comments below.

Several reviewers ask, essentially, “What does this complicated abstract definition buy us?” That’s obviously a fair question, and in any revision we’ll rewrite the introduction and conclusion to underline the goals:

1. Our motivation comes from interpretability work, where there has been a great deal of discussion of the existence of “world models” (See especially 2.3), but with an extremely informal and inconsistent set of definitions. We believe that this paper represents a step toward making many of these informal definitions more precise, while clearly excluding certain others (for instance, in the Vafa et al paper, a definition based purely on behavior rather than implementation).
2. One immediate benefit, for experimentalists, is that the diagram and nontriviality conditions give a checklist for experimental results, one positive and two negative. Positive: to say Z has a representation of a world model, show that a simple function g: Z → M can be learned. Negative conditions: to say this world model was learned by the network, show that no function of comparable simplicity can be learned X → M. To say that the learned world model is emergent, show that no function of comparable simplicity can be learned Y → M.

This framework is a definition, not an empirical result, so isn’t “falsifiable” as such. However, we believe that it will help make it easier to compare empirical work. For example, the Vafa et al. paper and the Li et al. paper describe seemingly inconsistent results for the same network (Othello-GPT). However, with this definition in hand it becomes clear that they are talking about two different things. A second example is the paper “Language Models Represent Space and Time” from Gurnee and Tegmark: this caused some controversy due to the fact that there was no test for whether the “world model” was present already in token embeddings. Our checklist of nontriviality conditions formalizes that concern. Given the comments here, we can add examples and make these benefits more explicit in the paper.

Reviewers also suggested fleshing out the relation between our definition and ideas around predictive coding or probabilistic modeling. This is an excellent point: our definition is focused specifically on representations of world state (“where are the walls”), rather than predicting the future (“will I bump into a wall”) or modeling the causal structure of the world (“If I had turned left, would I have bumped into the wall”). Indeed, our definition is meant to crisply focus on issues of representation, and separate these from issues around prediction or causal modeling. We will clarify this in any revision.

Finally, multiple reviewers mentioned that our usage of the word “understanding” is confusing. We see what you mean, and will clarify appropriately. In particular, we’ll include a targeted discussion in the introduction of how debates about “understanding” relate to world models, and then plan to avoid the word in the remainder of the paper.

---

### Note · Authors · 2025-01-21

I have read and agree with the venue's withdrawal policy on behalf of myself and my co-authors.